# Frameworks, measures, and interventions for HIV-related internalised stigma and stigma in healthcare and laws and policies: systematic review protocol

Susanne Hempel ,[1] Laura Ferguson,[2] Maria Bolshakova,[1] Sachi Yagyu,[1] Ning Fu,[3] Aneesa Motala,[1] Sofia Gruskin[2]

[1]Southern California Evidence Review Center, University of Southern California, Los Angeles, California, USA
[2]Institute on Inequalities in Global Health, University of Southern California, Los Angeles, California, USA
[3]Department of Economics, Shanghai University of Finance and Economics, Shanghai, China

**Correspondence to**
Dr Susanne Hempel;
susanne.hempel@med.usc.edu

## ABSTRACT

**Introduction** There is strong global commitment to eliminate HIV-related stigma. Wide variation exists in frameworks and measures, and many strategies to prevent, reduce or mitigate stigma have been proposed but critical factors determining success or failure remain elusive.

**Methods and analysis** Building on existing knowledge syntheses, we designed a systematic review to identify frameworks, measures and intervention evaluations aiming to address internalised stigma, stigma and discrimination in healthcare, and stigma and discrimination at the legal or policy level. The review addresses four key questions (KQ): KQ1: Which conceptual frameworks have been proposed to assess internal stigma, stigma and discrimination experienced in healthcare settings, and stigma and discrimination entrenched in national laws and policies? KQ2: Which measures of stigma have been proposed and what are their descriptive properties? KQ3: Which interventions have been evaluated that aimed to reduce these types of stigma and discrimination or mitigate their adverse effects and what are the effectiveness and unintended consequences? KQ4: What common 'critical factors for success or failure' can be identified across interventions that have been evaluated? We will search PubMed, PsycINFO, Web of Science, Universal Human Rights Index, HeinOnline, PAIS, HIV Legal Network, CDSR, Campbell Collaboration, PROSPERO and Open Science Framework. Critical appraisal will assess the source, processes and consensus finding for frameworks; COnsensus-based Standards for the selection of health Measurement Instruments criteria for measures; and risk of bias for interventions. Quality of evidence grading will apply . A gap analysis will provide targeted recommendations for future research. We will establish a compendium of frameworks, a comprehensive catalogue of available measures, and a synthesis of intervention characteristics to advance the science of HIV-related stigma.

**PROSPERO registration number** CRD42021249348.

## Strengths and limitations of this study

► Extensive literature searches will summarise HIV stigma evidence.
► This review will establish a comprehensive compendium of frameworks, a user-friendly catalogue of existing measures, and a clear synthesis of the effects of interventions.
► The review is limited to internalised stigma, stigma and discrimination in healthcare, and stigma and discrimination at the legal or policy level.

idea of the power dynamics that allow the distancing or othering so often associated with stigma, define stigma as 'the co-occurrence of labelling, stereotyping, separation, status loss and discrimination in a context in which power is exercised'.[2] Relf et al define stigma as 'a social phenomenon that occurs when a person is evaluated as having an undesirable trait, attribute or behaviour and is subsequently deemed imperfect by societal standards'.[3] Each of these definitions brings in something slightly different and many of the core concepts are neatly summed up by Parker and Aggleton who note that 'stigma functions at the intersection of culture, power and difference'.[4] Stigma can be described as a dynamic process of devaluation that significantly discredits an individual in the eyes of others, such as when certain attributes are seized on within particular cultures or settings and defined as discreditable or unworthy.

The term stigma is often used in the literature to encompass both stigma and discrimination even as these are conceptually distinct. While stigma refers to an attitude or belief, discrimination is the behaviour or action that results from those attitudes or beliefs. Hence, when stigma is acted on, the result can be discrimination. Discrimination may refer to any form of arbitrary distinction,

## INTRODUCTION

Stigma is derived from a Greek word meaning a mark or stain. Much work around HIV-related stigma uses as its starting point Goffman's 1963 definition as 'an attribute that is deeply discrediting'.[1] This has been furthered by Hatzenbuehler et al who, bringing in the

exclusion or restriction affecting a person, usually (but not only) because of an inherent personal characteristic or perceived membership of a particular group.

HIV-related stigma has been defined by the Joint United Nations Programme on HIV/AIDS (UNAIDS) as negative beliefs, feelings and attitudes towards people living with HIV, groups associated with people living with HIV (eg, families of people living with HIV) and other key populations at higher risk of HIV infection, such as people who inject drugs, sex workers, men who have sex with men and transgender people.[5] Although there is no universal consensus as to their categorisation, and many perceive these to be along a continuum, different domains have been identified within HIV-related stigma, including internalised, anticipated, perceived, enacted, externalised and structural stigma. Different types of stigmas can be experienced and assessed both alone and in combination, each of which is experienced differently and therefore must be addressed differently. Discrimination, as defined under international human rights law, is any distinction, exclusion or restriction based indirectly or directly on grounds prohibited under international law, which has the effect or intent of nullifying the recognition, enjoyment or exercise on an equal basis of others of all human rights and fundamental freedoms, in the political, economic, social, cultural, civil or any other field.[6] In the case of HIV, this can be discrimination based on a person's real or perceived HIV-positive status, irrespective of whether or not there is any justification for these measures.[7] It can rise to the level of a human rights violation. HIV-related discrimination is therefore any distinction, exclusion or restriction (sometimes referred to as acts or omissions) based indirectly or directly on a person's real or perceived HIV status.[8]

There is strong global commitment to eliminate HIV-related stigma, starting with global political commitments and reflected in global and national strategies as well as the multitude of organisations and collaborations working to address stigma.[9 10] Yet, learning across interventions designed to mitigate against the experience and harmful impacts of stigma is limited by the multitude of evaluation frameworks and measures in use to assess the different dimensions of stigma. For example, a recent review of interventions to address self-stigma did not include a formal meta-analysis due to the heterogeneity of measures used, with eight different scales used across 20 studies.[7] A 2015 UNAIDS report documented over 60 tools to assess and/or address stigma and discrimination within healthcare settings.[11] Assessment of stigma and discrimination entrenched in laws and policies also takes multiple forms including the People Living with HIV Stigma Index, Module 6 of the Demographic and Health Survey, the Integrated Bio-Behavioural Survey, legal environment assessments, the Global Fund baseline assessment methodologies, and the National Commitments and Policy Instrument of the Global AIDS Monitoring process.

Understanding the state of the research in relation to measurement of self-stigma, in accessing health services, and in laws and policies, is needed to help inform future efforts, at all levels, to better address stigma and support people's health and well-being. This review will systematically identify and assess frameworks, measures and interventions of HIV-related individual internalised stigma; both stigma and discrimination within healthcare settings; and stigma and discrimination at the legal and policy level. The intersectionality of HIV-related stigma with other forms of stigma such as stigma relating to 'key' and 'vulnerable' populations that are disproportionately affected by HIV is critical. The review will focus on HIV-related stigma itself, acknowledging as possible where intersectionality is addressed and, specifically, if there appear to be particular gaps in attention to stigma with respect to specific population groups.

### Review questions
The systematic review will be guided by four key questions.

#### Key question 1
Which conceptual frameworks have been proposed to assess internal stigma, stigma and discrimination experienced in healthcare settings, and stigma and discrimination entrenched in national laws and policies?

#### Key question 2
Which measures (eg, assessment scales) of these different types of stigma and discrimination have been proposed and what are their descriptive properties?

#### Key question 3
Which interventions have been evaluated that aimed to reduce these types of stigma and discrimination or mitigate their adverse effects and what are the effectiveness and unintended consequences of the interventions?

#### Key question 4
What common 'critical factors for success or failure' can be identified across the interventions that have been evaluated that might inform future interventions?

### METHODS AND ANALYSIS
The reporting of the protocol and the review follow the Preferred Reporting Items for Systematic Reviews and Meta-Analyses guidelines. The systematic review is part of a larger project undertaken by the International AIDS Society (IAS). It is accompanied by four national initiatives to explore stigma and discrimination in the local contexts of Kenya, Malawi, South Africa and Zambia through key informant interviews and grey literature searches. The project started in November 2020 and is planned to be completed by March 2022.

The systematic review will follow a transparent and rigorous procedure to minimise review selection and reporting bias. We will search multiple disciplinary and interdisciplinary sources to ensure all relevant studies

are captured. Citations and full text publications will be screened by independent literature reviewers to reduce reviewer errors and bias. Eligibility decisions, including reasons for exclusions, will be tracked in citation management software. Data abstraction and critical appraisal will be conducted in online software designed for systematic reviews using detailed, pilot-tested forms with clear reviewer instructions to avoid ambiguity and ensure replicability of coding decisions. The collected data will be accessible in a review data repository.[12]

## Context

Our systematic review will be embedded in the context of existing research syntheses. To date, a substantial number of systematic reviews has been published that address different aspects of HIV-related stigma and discrimination. These include associations of stigma (covariates, causes or effects),[13–40] access to care,[41–66] HIV testing,[67–91] country-specific explorations of stigma or discrimination,[92–109] the role of stigma and discrimination in treatment adherence,[110–127] HIV experiences of people living with HIV or their care givers,[128–141] stigma/discrimination reduction in the community,[142–153] HIV disclosure considerations,[154–162] stigma/discrimination in healthcare,[163–170] rights and regulations,[171–177] intersectionality,[178–183] measuring stigma,[184–189] HIV prevention,[190–193] stigma reduction in low income countries,[194–197] self-stigma[7 198 199] and other, unique topics.[200 201] We build on these reviews which outline existing research and point to persistent knowledge gaps. The reviews will be instrumental for reference-mining to ensure that all relevant material has been considered.

We will identify and categorise existing systematic reviews to explore the research field further. Systematic reviews will be identified through PubMed (biomedical literature) using the systematic review filter, through PsycINFO (psychosocial literature) and Web of science (general science literature including legal and policy analysis), as well as through the Cochrane Database of Systematic Reviews (focus on health) and the Campbell Collaboration (focus on social sciences). Furthermore, we will search the review registries PROSPERO and Open Science Framework to ensure that all relevant registered systematic reviews have been identified.

## Analytical framework

Given that there is no agreed nomenclature in this interdisciplinary field, we established working definitions of the concepts 'stigma,' 'internalised stigma' and 'discrimination' for the purpose of this systematic review:

► Stigma refers to beliefs and/or attitudes about HIV.
► Discrimination refers to the behaviours that result from attitudes or beliefs about HIV.
► Internalised stigma (self-stigma) refers to a person living with HIV internalising negative attitudes associated with HIV and accepting these as applicable to themselves.

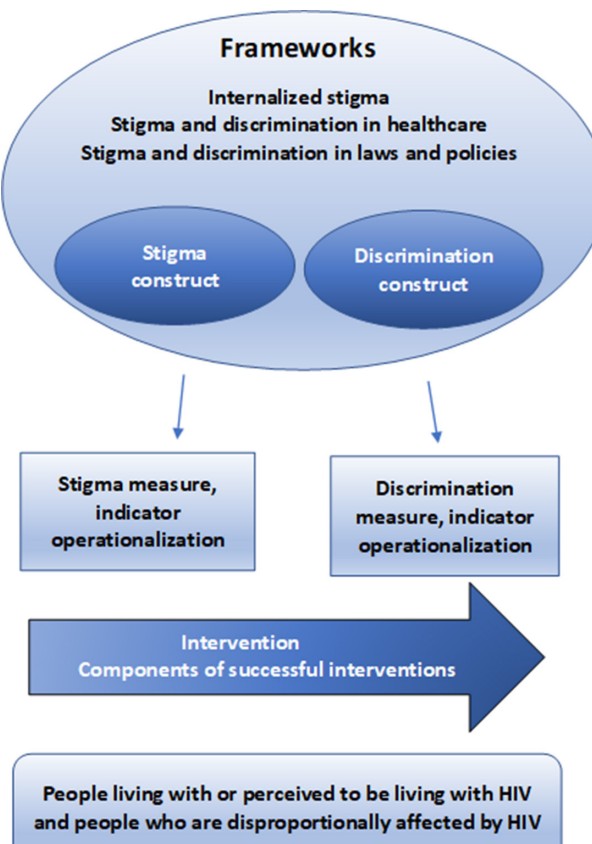

**Figure 1** Analytical framework.

► Stigma and discrimination in healthcare refers to negative beliefs and behaviours based on perceived or actual HIV status experienced in healthcare delivery settings.
► Stigma in laws and policies refers to distinctions, exclusion or restriction based on perceived HIV status or membership of a group that is vulnerable to HIV.

The analytical framework depicted in figure 1 maps the review's four Key Questions and outlines the review's content, that is, frameworks, measures and intervention evaluations, targeted in this systematic review.

The figure includes existing conceptual models for HIV-related stigma, including frameworks such as the global HIV stigma reduction framework,[202 203] and logic models addressing stigma and discrimination, potential causes or effects (key question 1). The frameworks comprise of the underlying construct of stigma and/or discrimination and their components. The analytical framework shows also the measures, indicators or operationalisations of the constructs stigma and discrimination that are used to measure and quantify stigma and discrimination (key question 2). The arrow represents interventions aiming to prevent, reduce, or mitigate stigma and discrimination (key question 3), as well as the 'critical success factors' of tested interventions (key question 4).

The systematic review is focused on people with living or perceived to be living with HIV and people who are disproportionately affected by HIV. For the purposes of

this review and based on the UNAIDS definition, the latter group includes gay men and other men who have sex with men, sex workers, transgender people, people who inject drugs and prisoners and other incarcerated people.[204]

### Search strategy

To identify primary research studies, we will search the health-specific research database PubMed, in particular to identify research on stigma experienced in healthcare settings. We will search PsycINFO to identify psychological and social research on stigma and use the general scientific research database Web of Science, in particular to identify legal and policy analyses on stigma and discrimination. We will identify government and nongovernmental organisation reports indexed in the Universal Human Rights Index, HeinOnline, PAIS and HIV Legal Network.

Additional grey literature searches will target organisations relevant to key populations affected by HIV and funders of stigma research. Specifically, we will search the websites of the IAS, UNAIDS, United Nations Development Programme, STRIVE, Health Policy Plus and Sage (resource-sharing community for Canadian HIV and hepatitis C service providers).

Targeted search strategies for each key question will combine free-text search terms with controlled vocabulary of the individual databases. Draft search strategies are shown in online supplemental appendix 1. The searches will be designed, executed and documented by an Evidence-based Practice Centre librarian. The searches outside of research databases will be instrumental in identifying conceptual models to assess internal stigma and stigma entrenched in national laws and policies. The multiple sources and interdisciplinary approach aim to reduce selection bias being introduced in the literature review.

### Eligibility criteria

We use a participant, independent variable, comparator or study design, outcome/measure, timing and setting framework to structure the eligibility criteria:

### Participants

People living with or perceived to be living with HIV and people from groups who are disproportionately affected by HIV infection. We will exclude studies of mixed populations and other participant targets unless the study provides HIV-relevant subgroup analyses.

### Independent variable

Frameworks will include models outlining and relating to multiple components of stigma and/or discrimination, including conceptual frameworks, logic models, taxonomies and analytic models for assessment, prevention, reduction or mitigation of stigma. Measures will include self and peer report measures used for formal assessment of stigma and discrimination. Studies using a published measure will be included if the psychometric

properties of the measure are a central focus of the study (eg, evaluating the predictive validity of an existing measure). Eligible intervention evaluations may test strategies and policies aimed at preventing, reducing or mitigating HIV-related stigma and discrimination. Intervention evaluations aimed at self-stigma, stigma and discrimination in healthcare, and stigma and discrimination at the legal or policy level are eligible. We will exclude publications exclusively addressing community and cultural stigma.

### Comparator/study design

Publications introducing frameworks will be included regardless of the comparator or study design. Measure research needs to describe the tool in sufficient detail to be included but needs no comparator. Experimental evaluations of interventions have to report on a concurrent comparator (eg, randomised controlled trial, controlled clinical trial, comparative studies, quasi-experimental studies, natural experiments) to be eligible. Observational evaluations of healthcare interventions not under the control of the investigator and influenced by secular trends, have to report on a historic or concurrent comparator (eg, pre–post, cohort study comparing two cohorts) and be sufficiently large or encompassing to be eligible (either demonstrating statistical power in a power analysis, reporting on large-scale evaluations of 200 participants or more, or ensuring that all eligible units have been targeted such as all healthcare providers in a hospital received the intervention). For studies evaluating the impacts of laws, no comparator is required if the study author can demonstrate an alternative analysis of determining the effect of the law not compared with a prior period or different legislative environment. Systematic reviews will be retained for an umbrella review providing context and for reference-mining.

### Outcome

Framework publications will be included regardless of any reported outcomes. Measure research needs to report a detailed description of the measure, the development process, or the evaluation or validation of the measure. Intervention research needs to provide a structured evaluation of an indicator of stigma or operationalisation of discrimination to be eligible.

### Timing

Frameworks will not be restricted by publication year. For measures and interventions, only those published from 2008 on will be included, building on the first People Living with HIV Stigma Index published in 2008, which transformed thinking around HIV-related stigma measurement.[205]

### Setting

The review is not restricted regarding setting but we will restrict to English language publications.

 Hempel S, et al. BMJ Open 2021;11:e053608. doi:10.1136/bmjopen-2021-053608

## Data abstraction

For the frameworks, we will abstract the author group and publisher; publication year, scope, aim or purpose of the framework; subtype and definition of the constructs stigma and/or discrimination; addressed targets (eg, people living with HIV and their families); framework components; and a broad summary of the framework. The Draft Evidence Table Key Question 1 will summarise the identified frameworks (see online supplemental appendix 1).

For measures, we will document the author group; publication year; name of the tool; the stigma or discrimination subtype being assessed, the underlying framework (where applicable) and definitions of stigma and discrimination; the targeted population; the surveyed population used to develop or assess the measure; the scale structure of the tool, number of items, and answer mode; the documented reliability; and evidence of validity. Draft Evidence Table Key Question 2 outlines the evidence table documenting this information for each included publication (see online supplemental appendix 1).

For intervention evaluations (key question 3), we will document the study identification details, year of publication, country, study design, sample size, participant details, context or setting; intervention type, intervention description and intervention components, comparator type and comparator description, definition and measures of stigma and discrimination, and findings for the outcome measures. In order to document the effect of the intervention concisely, we will abstract data for the main effectiveness signal and adverse events or unintended consequences. In addition, we will record any information provided by the study authors on the appropriateness of the used measures, intervention or comparator. The draft evidence table is shown in online supplemental appendix 1 (Draft Evidence Table Key Question 3).

## Critical appraisal and analysis plan

For frameworks, we will assess the source (eg, published by an individual author group or endorsement by a professional organisation) and processes used to develop the framework, including stakeholder involvement and formal consensus finding methods.[206 207]

For measures, we will evaluate the demonstrated reliability (internal consistency, test–retest stability, rater agreement) as well as evidence of the validity (eg, content, criterion or construct validity) applying COnsensus-based Standards for the selection of health Measurement Instruments criteria.[208]

Intervention evaluations will be evaluated for potential selection, detection, performance, attrition, reporting and study-specific sources of bias, adapting RoB 2 and ROBINS-I criteria.[209 210]

To determine the effects of the interventions, we will compute measure-independent effect estimates for all included studies, that is, standardised mean differences for continuous outcomes and relative risks for categorical

outcomes, to facilitate comparisons across studies. Interpretation of the effect sizes will take the statistical significance of the difference compared with the control or preintervention status into account as well as the statistical power of the study to detect an effect. Where possible, results of studies across interventions will be summarised in random effects meta-analyses applying Hartung-Knapp corrections for small samples.[211]

To explore which interventions are successful and to determine what characterises these interventions, we will first broadly categorise the type of intervention. The categorisation will be undertaken based on the abstracted intervention description and will be blind to the findings of the study. The categories will be used to identify subgroups of more homogeneous intervention types. In a second step, we will derive a set of common intervention components drawn from the identified literature and informed by existing intervention frameworks and taxonomies. The presence or absence of components will be documented for each successful and each unsuccessful intervention, together with context information such as the country of evaluation. The draft component tables for the successful and unsuccessful interventions are shown in online supplemental appendix 1 (Draft Component Table Key Question 4a, Draft Component Table Key Question 4b).

Categorising the evaluations as successful or unsuccessful will follow a transparent algorithm and will be determined by two independent literature reviewers to reduce errors and bias. For all studies, the reduction of stigma will be determined. Assessments will be based on the results in the intervention group relative to a control group. In the absence of a concurrent control group, the change compared with the preintervention status will be used to determine intervention success. Categorising interventions as successful or unsuccessful will qualitatively stratify the identified research. We acknowledge that this step will lose the granularity of a continuous variable. However, we anticipate that a broad categorisation is necessary given the diversity of the approaches. We will use transparent methods to categorise studies and discrepancies in the effect classification between reviewer ratings will be resolved through discussion in the review team.

Determining characteristics of successful interventions and success factors in often complex and multicomponent interventions, as well as reasons for failure, will require a careful analysis of the components or 'active ingredients' of the interventions. We will apply principles of qualitative comparative analysis and review the established component matrix across the successful interventions in seeking to determine which individual and figurations of factors appear to be associated with success.[212] We will explore differences in the component structure between interventions that were determined to be successful and those that were determined to be unsuccessful to identify components likely be associated with success. We will conduct meta-regressions across studies to confirm effects

of components. Meta-regressions will add the component of interest to the meta-analytic model to determine whether its presence or absence affects the size of the intervention effect.

## Summary of findings and body of evidence assessment

We will document transparent criteria to evaluate the certainty in the evidence across the included research. We will adapt the eight Grading of Recommendations, Assessment, Development and Evaluation criteria study limitation, inconsistency, imprecision, indirectness, reporting bias to upgrade, and the criteria large effect, dose–response relationship, and confounding would mask an effect, to downgrade the quality of evidence.[213] While the criterion study limitation is applicable to all key questions, inconsistency (eg, of reliability estimates across studies) will be used to interpret the quality of evidence for key questions 2, 3 and 4, and the criterion imprecision (eg, range of reported reliability estimates) will be applied to key questions 2 and 3. All other criteria will primarily apply to key question 3 and the evidence statements that can be formulated to answer this key question.

The evaluation of the body of evidence will be used to arrive at internationally accepted certainty categories that communicate our confidence in the findings using the categories high, moderate, low and very low. High indicates confidence that the true effect is similar to the estimated effect, moderate indicates the true effect is probably close to the estimated effect, low suggests the true effect might be markedly different from the estimated effect, and very low signals that the true effect is probably markedly different from the estimated effect. We will review the appropriateness of the starting point of low quality of evidence for non-randomised studies before upgrading or downgrading of the evidence to avoid floor effects and ensure meaningful differentiation.

## Synthesis

The identified evidence will be documented in comprehensive tables and figures. The literature flow will be documented in a literature flow diagram and account for all identified research. Critical appraisal of frameworks and measures will be integrated into the evidence tables. The risk of bias across intervention studies will be documented in a risk of bias figure showing the distribution for each criterion. All included research will be documented in concise evidence tables providing details for each included framework, measure and intervention evaluation (see drafts in online supplemental appendix 1).

With this review, we intend to establish a compendium of existing frameworks for HIV-related stigma and discrimination. Frameworks published under Creative Commons licences, and where copyright agreements allow these for other frameworks, will be shown in full to allow a meaningful overview. Where copyright assertions cannot be obtained, the framework evidence table will include the link to the framework if available in the public domain.

The evidence table for measures aims to provide a resource for future HIV stigma and discrimination research. It will provide a comprehensive overview of the available measures to help select tools for future studies.

The evidence table and component tables for the identified interventions aim to document the existing knowledge base for interventions to address self-stigma, stigma and discrimination in healthcare, and stigma and discrimination in laws and policies. A detailed exploration of intervention components aims to support efforts in prevention, reducing and mitigating HIV-related stigma and discrimination.

In addition to a narrative synthesis, we will document the evidence across research in summary of findings tables with one table for each key question. The table summarising key question 1 will document the number and type of the identified stigma and discrimination frameworks. The summary of findings table summarising key question 2 will document the number and type of identified stigma and discrimination measures, listing the measure details and reliability (evidence that the tool is measuring accurately and consistently) and validity (evidence that the tool is measuring what it is supposed to measure). The third table will document the evidence for the interventions (key question 3), organised by context and modality (eg, healthcare facility intervention, law or policy change) and within by outcome category (effectiveness, unintended consequences), together with the quality of evidence assessment. The fourth summary of findings table will document commonalities of successful interventions and failed interventions (key question 4).

## Gap analysis

Part of the systematic review will be a formal gap analysis. We will use a structured approach to document research gaps. Gaps will be documented in a Study design, Participants, Interventions, Framework, Outcomes, Context/ Country framework, with attention to other relevant categories that may emerge during the evidence review. The gap analysis will document research needs and provide concrete recommendations for future research.

## Patient and public involvement statement

The draft protocol was peer-reviewed by international content experts and a representative of the community of persons living with HIV to ensure that the review asks the right questions. Stakeholders will be asked to review the draft review to ensure that all relevant frameworks, measures and intervention evaluations have been captured and that the review contributes meaningfully to the knowledge base and to ensure that the evidence review is as impactful as possible.

## Ethics and dissemination

The systematic review conduct will be transparent and is designed to advance research to support people living with or suspected to be living with HIV and people disproportionally affected by HIV. The review is considered

exempt as it does not involve human subjects and does not require review by the human subject protection committee.

The review will be registered in PROSPERO, the results will be published in a journal manuscript and presented at research conferences. The data will be made available in the Systematic Review Data Repository.[12]

**Acknowledgements** We thank Lucy Stackpool-Moore, Tessa Oraro-Lawrence, Kasoka Kasoka, Brent Allan, and members of the country team for helpful comments.

**Contributors** SH, LF and SG developed the systematic review, all authors (SH, LF, MB, SY, NF, AM and SG) contributed to this protocol. SH serves as guarantor and accepts full responsibility for the work and the conduct of the study, had access to data and controlled the decision to publish. The corresponding author attests that all listed authors meet authorship criteria and that no others meeting the criteria have been omitted.

**Funding** This work (Global Systematic Evidence Review: Getting to the Heart of Stigma) is supported by the International AIDS Society (IAS). This work was supported, in whole or in part, by the Bill & Melinda Gates Foundation [Grant Number INV-004364]. Under the grant conditions of the Foundation, a Creative Commons Attribution 4.0 Generic License has already been assigned to the Author Accepted Manuscript version that might arise from this submission.

**Disclaimer** The funder had no role in the development of the protocol or decision to submit this manuscript.

**Competing interests** None declared.

**Patient consent for publication** Not applicable.

**Provenance and peer review** Not commissioned; externally peer reviewed.

**Data availability statement** The data will be made available in the Systematic Review Data Repository (https://srdr.ahrq.gov/).

**ORCID iD**
Susanne Hempel http://orcid.org/0000-0003-1597-5110

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
