## [Reviewer comments · BMJ Open]

ARTICLE DETAILS

TITLE (PROVISIONAL)	Frameworks, measures, and interventions for HIV-related internalized stigma and stigma in healthcare and laws and policies: Systematic Review Protocol
AUTHORS	Hempel, Susanne; Ferguson, Laura; Bolshakova, Maria; Yagyu, Sachi; Fu, Ning; Motala, Aneesa; Gruskin, Sofia

VERSION 1 – REVIEW

REVIEWER	Broady, Timothy UNSW, Centre for Social Research in Health
REVIEW RETURNED	29-Jun-2021

GENERAL COMMENTS	This is a very thorough and clearly written protocol for a systematic review, with a clear rationale for its conduct. I have some very minor points to clarify: Throughout the introduction, there is a lack of references. The content outlines the background and rationale for the study very clearly, but additional referencing is needed. What is the expected timeframe for the conduct of the systematic review? I note that frameworks will not be restricted by publication year, but measures and interventions will only be included if published from 2008 onwards. As estimated date range for the review to be undertaken will help understand the full time period of publication that will be included. Critical appraisal and analysis plan: It would be helpful to further clarify how categories will be determined in relation to determining which interventions are successful and what characterizes these interventions. Similarly, please clarify if the goal is to categorise evaluations into a dichotomous “successful” vs. “unsuccessful” framework. While these processes will be influenced by the results of previous steps described in the paper, some further clarification around the rationale and process of this appraisal is justified.
--

REVIEWER	Montess, Michael University of Victoria, School of Public Health & Social Policy
REVIEW RETURNED	04-Aug-2021

GENERAL COMMENTS	The authors have produced a well-planned and detailed systemic review protocol to study an important topic in the literature on HIV stigma. Since there are many different evaluation frameworks presently being used to understand HIV stigma and discrimination this protocol is an important step in understanding how current frameworks either succeed and fail, specifically for people living with or suspected to be living with HIV and people disproportionately affected by HIV. In turn, this will produce a helpful tool for preventing, reducing, or mitigating different kinds of HIV
---

	stigma, including internalized stigma, stigma and discrimination in healthcare, and stigma and discrimination in law or policy. While the authors have provided a sound analytic framework consisting of working definitions of the key terms of “stigma” and “discrimination”, they could also draw on some of the more theoretical literature on both of these terms, especially in the introduction section of the protocol or in future work on the study, in order to provide an even stronger foundation for their definitions. Citations or references regarding other definitions of these terms could also help better situate their working definitions in the literature. Nevertheless, as HIV research is an interdisciplinary field, nomenclature varies and the authors have already contributed to the literature by presenting these reasonable working definitions and writing this promising and meticulous systemic review protocol.
--	---

VERSION 1 – AUTHOR RESPONSE

Reviewer: 1

Dr. Timothy Broady, UNSW

Comments to the Author:

This is a very thorough and clearly written protocol for a systematic review, with a clear rationale for its conduct. I have some very minor points to clarify:

Thank you for the positive feedback.

Throughout the introduction, there is a lack of references. The content outlines the background and rationale for the study very clearly, but additional referencing is needed.

Thank you for noting this. We have added more references to the introduction to better ground its content in existing literature.

What is the expected timeframe for the conduct of the systematic review?

I note that frameworks will not be restricted by publication year, but measures and interventions will only be included if published from 2008 onwards. As estimated date range for the review to be undertaken will help understand the full time period of publication that will be included.

We have added this information to the method section.

Critical appraisal and analysis plan: It would be helpful to further clarify how categories will be determined in relation to determining which interventions are successful and what characterizes these interventions. Similarly, please clarify if the goal is to categorise evaluations into a dichotomous “successful” vs. “unsuccessful” framework. While these processes will be influenced by the results of previous steps described in the paper, some further clarification around the rationale and process of this appraisal is justified.

You are raising an important point and while a necessary step, it is a challenge in practice and there are clear limitations of the methodology. We have revised the section, added more detail, and have acknowledged the limitation.

Reviewer: 2

Dr. Michael Montess, University of Victoria

Comments to the Author:

The authors have produced a well-planned and detailed systemic review protocol to study an important topic in the literature on HIV stigma. Since there are many different evaluation frameworks presently being used to understand HIV stigma and discrimination this protocol is an important step in understanding how current frameworks either succeed and fail, specifically for people living with or suspected to be living with HIV and people disproportionately affected by HIV. In turn, this will produce a helpful tool for preventing, reducing, or mitigating different kinds of HIV stigma, including internalized stigma, stigma and discrimination in healthcare, and stigma and discrimination in law or policy.

Thank you.

While the authors have provided a sound analytic framework consisting of working definitions of the key terms of “stigma” and “discrimination”, they could also draw on some of the more theoretical literature on both of these terms, especially in the introduction section of the protocol or in future work on the study, in order to provide an even stronger foundation for their definitions. Citations or references regarding other definitions of these terms could also help better situate their working definitions in the literature. Nevertheless, as HIV research is an interdisciplinary field, nomenclature varies and the authors have already contributed to the literature by presenting these reasonable working definitions and writing this promising and meticulous systemic review protocol.

Thank you for drawing our attention to this. We have added text to present different definitions grounded in current literature to better contextualize the definitions that we use throughout.